# Assessment of Potential Probiotic and Synbiotic Properties of Lactic Acid Bacteria Grown In Vitro with Starch-Based Soluble Corn Fiber or Inulin

**DOI:** 10.3390/foods11244020

**Published:** 2022-12-13

**Authors:** Iris M. García-Núñez, Arlette Santacruz, Sergio O. Serna-Saldívar, Sandra L. Castillo Hernandez, Carlos A. Amaya Guerra

**Affiliations:** 1Facultad de Ciencias Biológicas, Universidad Autónoma de Nuevo León, Ciudad Universitaria, Pedro de Alba s/n, San Nicolás de los Garza 66451, Mexico; 2Centro de Biotecnología FEMSA, Escuela de Biotecnología y Alimentos, Tecnológico de Monterrey-Campus Monterrey, Av. Eugenio Garza Sada 2501 Sur, Monterrey 64849, Mexico

**Keywords:** probiotics metabolism, prebiotics fermentation, synbiotics, starch-based soluble corn fiber, short chain fatty acids

## Abstract

This research is aimed to search for suitable probiotic plus prebiotic combinations for food applications. Sixteen bacteria were tested for resistance to low pH, bile salts and antibiotics, and their adhesion to Caco-2 cells, in order to select potential probiotics. Then, two bacteria were selected to study short chain fatty acids production in a starch-based soluble corn fiber or inulin media. *Lactiplantibacillus plantarum* V3 and *L. acidophilus* La3 manifested the best probiotic features with a remarkable adhesion ability (23.9% and 17.3%, respectively). Structural differences between fibers have an impact on how each one is metabolized, both in their capacity of being easily fermented and in the short chain fatty acids profile obtained: *L. acidophilus* La3 in inulin fermentation yielded the highest total short chain fatty acids (85.7 mMol/L), and, in starch-based soluble corn fiber fermentation, yielded the highest butyric acid content (0.31 mMol/L). This study provides valuable information for future design of synbiotics for food applications.

## 1. Introduction

In 2001, the U.S. Food and Drug Administration and the World Health Organization jointly defined the term probiotic and, more recently, in 2014, the International Scientific Association for Probiotics and Prebiotics revised the definition and updated it to “live microorganisms that, when administered in adequate amounts, confer a health benefit on the host” [1]. Among the most notable effects, the strains of probiotic bacteria should positively influence the composition of the intestinal microbiota and modulate the host’s immune system [2]. Probiotic microorganisms had to meet some criteria to ensure their delivery in sufficient amounts in their action site so they could exert beneficial effects, then, they must undergo and survive under very harsh conditions before reaching the lower intestinal tract. First, these microorganisms have to resist the acidic conditions of the stomach, where probiotics commonly reside 90 min in gastric acids at pH 2.5. Thereafter, they enter the upper intestinal tract and remain for approximately 120 min, where bile acids concentrations vary between 0.3 to 2% [3], respectively. Once they reach the lower large intestine, probiotics must be able to adhere to the epithelium in order to survive, colonize and grow and, as a result, provide health benefits to the host. Antibiotics are one of most important advances in modern medicine; however, their indiscriminate use in human and veterinary medicine have led to the increased spread of antibiotic resistance, which has diminished their effectiveness against infectious human diseases. Unfortunately, antibiotic resistance is reported among bacteria, including probiotic microorganisms. Therefore, it is recommended to determine antibiotic resistance profiles of probiotic bacteria, especially in terms of avoiding transmission of genes that encode resistance to clinically used drugs. Currently, many bacteria pure cultures are available for food and pharmaceutical applications. Nevertheless, the probiotic properties, functionality and benefits differ among strains. Consequently, they cannot be extrapolated even within bacteria belonging to the same species [4]. *Bifidobacterium* and lactic acid bacteria (LAB), such as *Lactobacillus*, are the most relevant and consumed probiotics in many food applications. LAB are commonly exploited for the bio-preservation of diverse foods and beverages, including many traditional fermented cereal foods like pozol, kenkey, mawe, boza, mageu, munkoyo and uji [5]. Many bacteria from collections and some industrial starter cultures have been tested to evaluate them as probiotic candidates, for example, *Lactiplantibacillus plantarum* DSM 9843 (299v) [6], *Lactiplantibacillus plantarum* ATCC 14917 [7], *Lactobacillus delbrueckii* ATCC 11842 [8], *Lacticaseibacillus casei* ATCC 393 [9], *Lactobacillus acidophilus* ATCC 4356 [10], *Lactococcus lactis* ATCC 11454 [11] and *Lacticaseibacillus rhamnosus* ATCC 53103 (GG) [12]. However, due to important differences in the methodology used to evaluate them, a direct comparison between bacteria is difficult.

A prebiotic is a substrate that is selectively utilized by host microorganisms conferring a health benefit. The intake of prebiotics can modulate to a large extent the colon microbiota by increasing the number of specific probiotic bacteria resulting in the change of the population dynamics of the microbiota [13]. Other proven health related effects of prebiotics are increased absorption of nutrients in the intestine, increasing of immunity, inhibition of carcinogenic toxicity, preservation of intestine membrane integrity, glycemic index and body weight decrease [14]. Most traditional and well-studied prebiotics are inulin, fructo-oligosaccharides and galacto-oligosaccharides; however, there are many other food fibers with prebiotic potential that have not been studied in detail. For example, starch-based corn fibers are novel ingredients in food products but they have not been thoroughly studied in terms of their prebiotic function. Starch-based soluble corn fiber (SCF) is a well-known maize-derived source of dietary fiber with potential selective fermentation properties. It can stimulate *Bifidobacteria* numbers in the overall gut microbiota [15] and in vitro tests have demonstrated it is well-fermented in the distal colon and leads to positive effects on the gut microbiota’s activity and composition [16].

The term synbiotic refers to “a mixture comprising live microorganisms and substrate(s) selectively utilized by host microorganisms that confers a health benefit on the host” [17]. An appropriate combination of both components in a single product should ensure a superior effect compared to the activity of the probiotic or prebiotic alone. Most popular combinations in synbiotic products are *Bifidobacterium* and/or *Lactobacillus* with fructo-oligosaccharides or inulin. Nonetheless, synbiotics application for the prevention of gastrointestinal diseases and modulation of intestinal microbiota in humans seems very promising in regards to testing new prebiotic–probiotic combinations. As mentioned before, among prebiotic fibers starch-based SCF has not been the object of many studies, thus, testing its capacity to support growth of specific probiotics could be promising.

The aim of this research was to select combinations of bacteria and starch-based soluble corn fiber or inulin as a first step to find new synbiotics, considering the resistance of the bacteria to gastric and intestinal fluids, adherence to intestinal epithelium, antibiotic susceptibility and generation of short chain fatty acids.

## 2. Materials and Methods

### 2.1. Bacterial Strains and Fibers

Table 1 contains the bacteria strains examined (and their suppliers) which are six starter cultures (SACCO) and ten additional culture collection strains (including two recognized probiotic bacteria (*Lactiplantibacillus plantarum* 299v and *Lacticaseibacillus rhamnosus* GG) that were employed as controls). Fibers used in fermentation experiments were highly soluble inulin (Orafti^®^ HSI, Beneo, Belgium) and starch-based soluble corn fiber (SCF) (PROMITOR™ Soluble Corn Fiber 85, Tate & Lyle, IL, USA), which is a glucose polymer obtained from partially hydrolyzed starch with a structure composed by a mixture of α 1-6, α 1-4 and α 1-2 glucosidic linkages [18].

### 2.2. Culture Conditions

In order to harvest the bacteria, lyophilized strains were grown three times in MRS broth (Difco, Sparks, MD, USA) at 37 °C for 18 h, whereas frozen conserved cultures were grown twice under the same conditions. LAB were stored at −80 °C in phosphate-buffer saline (PBS) solution with 20% glycerol. Before each test, bacteria were activated and grown in MRS broth for 18 h at 37 °C.

### 2.3. Evaluation of Resistance to Simulated Gastric and Intestinal Fluids

All bacteria strains were tested for in vitro gastric and intestinal fluids resistance according to the methods of Guan [3] with slight modifications. To summarize, bacteria were grown for 18 h at 37 °C, then, cells were harvested by centrifugation at 8960× *g* for 5 min and washed twice in sterile saline solution (0.5%). Cells were suspended in saline solution to adjust turbidity to McFarland No. 5 (approximately 10^9^ CFU/mL). Culture suspensions were properly diluted to obtain initial culture counts. The numbers of colony forming units were determined by MRS agar plate count. In order to evaluate gastric fluid resistance, 200 µL of culture suspension of bacteria was inoculated in 1800 µL of simulated gastric solution (0.5% NaCl, 0.3% pepsin (from porcine gastric mucosa; Sigma-Aldrich, St. Louis, MO, USA) adjusted to pH 2.5). The bacteria’s survival was examined after 1.5 h of incubation at 37 °C and colony-forming units were determined under the aforementioned conditions. On the other hand, the simulated intestinal solution consisted of 0.5% NaCl, 0.2% pancreatin (from porcine pancreas; Sigma-Aldrich, MO, USA) and 0.5% bile salts (MCD Lab, Edo de Méx., Mexico) at pH 8.0. A 200 µL volume of culture suspension of test bacteria was first inoculated in 1800 µL of simulated intestinal solution and incubated for 2 h at 37 °C, then the bacteria’s survival was determined by plate counting. The bacteria reductions were obtained by comparing counts before and after incubation at gastric or intestinal simulated fluids.

### 2.4. Evaluation of the Antibiotic Resistance

For later tests, the six most resistant bacterial strains were selected according to the survival results after exposure to gastric and intestinal fluids. The disk diffusion method according to CLSI guidelines [19] was followed in order to assess antibiotics resistance. Active cultures were streak-plated in MRS agar and incubated for 24 h at 37 °C. Then, colonies were taken and placed in PBS until turbidity matched the 0.5 McFarland standard (approximately 1.5 × 10^8^ CFU/mL). Once matched, the standardized culture was immediately inoculated on MRS agar by swabbing in three directions, then a multi-antibiotic disk (Bio-Rad S.A., Mexico) was placed. Antibiotic disks contained: Cephalotin, 30 µg; Erythromycin, 15 µg; Trimethoprim + sulfamethoxazole, 25 µg; Ceftriaxone, 30 µg; Netilmicin, 30 µg; Chloramphenicol, 30 µg; Ampicillin, 10 µg; Enoxacin, 10 µg; Penicillin, 10 µg; Amikacin, 30 µg; Gentamicin, 10 µg; or Dicloxacilin, 1 µg. Plates with antibiotic disks were incubated at 37 °C under anaerobic conditions. After 24 h, diameters of inhibition zones were measured. *Staphylococcus aureus* ATCC 25923 was streak plated on Mueller-Hinton agar and incubated for 18 h, it was included in tests as positive reference or control following the same steps. The diameters in mm of the inhibition zones were rated as susceptible, intermediate or resistant, according to classification find tables previously reported for *Lactobacillus* [20]. The interpretation of *Staphylococcus aureus* results was performed according to breakpoint tables published in CLSI guidelines [19].

### 2.5. Capacity of Bacterial Adhesion to Caco-2 Cells

Experiments to evaluate the adhesion of *Lactobacillus* strains were conducted by the adhesion assay method described by [21]. The Caco-2 cell line, purchased from the American Type Culture Collection, was grown in culture plates containing DMEM-F12 medium with 5% fetal bovine serum (FBS) at 37 °C, 5% CO2 and 90% relative humidity. A confluence of 90% was reached 1 week after seeding, cells were counted in order to inoculate 24-well tissue culture plates in concentration of 1 × 10^5^ cells per well. After confluence, the medium was maintained for two additional weeks prior to use in adhesion assays; during this time, DMEM-F12 medium was changed every other day. The bacterial active culture previously described was prepared by centrifuging at 8960× *g* for 5 min and washed twice with phosphate saline solution. Finally, bacteria were suspended in Dulbecco modified 4.5 pH Eagle medium (DMEM)–F-12 (Invitrogen). For the adhesion assay, the medium was removed first, then each well was inoculated with 0.5 mL of the bacterial culture mentioned above. Cells were allowed to incubate at 37 °C for 1.5 h to mediate adherence. The cells were then rinsed twice with PBS (prewarmed to 37 °C) and treated with 1% triton X-100 (Sigma, USA). Next, 0.2 mL of cell suspension was taken and a set of serial dilutions was prepared and plated out on MRS agar. Following incubation at 37 °C for 48 h, colonies were evaluated by plate counting. The percentage of adhesion ratio was calculated by comparing the total number of colonies counted after adhesion to the one in the suspension added to the tissue culture plate wells. *Salmonella enterica* ATCC 13311 was included in the adhesion test as a control. Three independent experiments were performed in which all strains were tested in triplicate.

### 2.6. Prebiotic Fermentation by Potential Probiotics

Two strains that exhibited the best performance in previous tests were selected to assess their capacity to ferment inulin or starch-based soluble corn fiber. MRS broths based on dextrose (control), inulin or starch-based SCF fermentations were tested. Culture media with prebiotics were made considering the fiber purity and residual sugar content claimed by manufacturer in order to test same concentrations of these components in both broths. The inulin and starch-based SCF purities were 88% and 87.5%, respectively, and contained sugar concentrations of 12 and 2%, respectively. Final sugar contents in all media were 2.4%. Due to its lower sugar content, the soluble corn fiber preparation was supplemented with dextrose to match the sugar concentration of the inulin counterpart. Fiber concentrations in treatments were 17.6% and 17.5% for inulin and starch-based SCF respectively, this formulation was made to appreciate the death phase in bacterial growth curves in control medium and to appreciate fibers fermentation after sugar consumption in inulin and starch-based SCF.

Taking into consideration that prebiotics are fermented by probiotics after passing through the gastrointestinal tract, all MRS broths were subjected to simulated digestion, as previously reported [22], with slight modifications. First, 250 mL of fresh prepared broths was heated to 95 °C, followed by equilibration to 25 °C and adjustment of the pH to 6.9 with 3N HCl. Thereafter, 65 mg of α-amylase (from porcine pancreas; Sigma-Aldrich MO, USA) was added and broths were incubated at 37 °C with agitation at 3× *g* for 5 min. Then, the pH was adjusted to 2.0 with 6 N HCl in preparation for the addition of 125 mg of pepsin (from porcine gastric mucosa; Sigma-Aldrich MO, USA). The contents were incubated at 37 °C with agitation at 3× *g* for 30 min. Finally, the pH was readjusted to 5.3 with 6 N NaOH followed by the addition of 125 mg of pancreatin (from porcine pancreas; Sigma-Aldrich MO, USA) and incubation for 30 min at 37 °C at 3× *g*. After that, the broths were autoclaved to avoid bacterial fermentation other than the strain to prove.

Broths were inoculated with *Lactiplantibacillus plantarum* V3 or *L. acidophilus* La3 at approximately 3 × 10^4^ CFU/mL and incubated for 48 h at 37 °C in a vacuum oven (VWR, Edo. de Méx., Mexico) to promote fermentation under anaerobic conditions. Aliquots were sampled at 0, 4, 8, 12, 20, 28, 36 and 48 h and immediately tested for titratable acidity (expressed as lactic acid equivalents), °Brix and pH. An aliquot of each sample was stored at −20 °C. Bacteria concentration during fermentation was quantified by quantitative real-time PCR. The Dneasy UltraClean Microbial Kit (Qiagen, CDMX, Mexico) was used to extract DNA from 1.8 g of fermented broth sample according to the manufacturer instructions. Cells were lysed briefly through a combination of mechanical force, heat and detergent, and then bound to a silica spin filter and then washed [23]. The resulting DNA was recovered in 50 µL Tris buffer.

Primers LacF (sequence 5′-GGAAACAGRTGCTAATACCG-3′) and LacR (sequence 5′-CACCGCTACACATGGAG-3′) (Eurofins MWG Operon, KY, USA), which recognize the 16S rRNA of *Lactobacillus* genus [24], were used. The amplification reactions were conducted with SYBR Green master mix (Applied Biosystems, CDMX, Mexico) and both primers (each at 500 nM concentration) had purified DNA and molecular water. Amplification (10 min at 95 °C, followed by 35 cycles of 10 s at 95 °C, 15 s at 61 °C and 20 s at 72 °C) and detection were performed on a Rotor-Gene 3000 (Corbett Research). The amount of genomic DNA extracted was determined by ultraviolet spectrophotometry at 260 nm (NanoDrop, Thermo Fisher, Monterrey, Mexico) and purity was verified by agarose gel electrophoresis and ethidium bromide staining.

### 2.7. Short and Medium Chain Fatty Acids Formation

Gas chromatography was used to identify if selected bacteria were capable of producing short chain fatty acids (SCFA) and medium chain fatty acids (MCFA) during fermentation as previously reported [25]. In order to purify the media, the fermented culture media were first centrifuged at 14,000× *g* at 4 °C for 10 min, then filtered through pore 0.45 µm hydrophilic PTFE syringe filters (Shimadzu CA, USA). Chromatographic analysis was made using an Agilent 6850GC system equipped with a flame ionization detector (FID) and an automatic liquid sampler (Agilent, CA, USA). A nitroterephthalic acid modified polyethylene glycol (HP-FFAP, J&W Scientific, Agilent Technologies Inc., CA, USA) of 25 m × 0.32 mm i.d. coated with 0.50 μm film thickness column was used. Helium was supplied as the carrier gas at a flow rate of 1.0 mL/min. The initial oven temperature program was 120 °C, maintained for 2.0 min, raised to 240 °C at 15 °C/min and held for 5.0 min. Glass wool (Supelco) was inserted in the glass liner of the split (5:1 ratio) injection port. The temperatures of the FID and the injection ports were 300 and 240 °C, respectively. The flow rates of hydrogen, air and helium as makeup gas were 40, 400 and 14 mL/min, respectively. The injected sample volume was 1 μL and the run time for each analysis was 20.7 min. An OpenLAB CDSChemStation software (Agilent, CA, USA) was utilized for data handling. An analytical standard solution mix (Supelco, Edo. de Mex., Mexico) containing formic, acetic, propionic, isobutyric, butyric, isovaleric, valeric, isocaproic, hexanoic and n-heptanoic acids at 10 mM concentration each was employed to identify SCFA on chromatograms based on their specific retention times. The stock standard solution was stored at 4 °C until used. Individual calibration curves were obtained for each compound through the standard SCFA mixture. From the chromatograms, the area under the peak was plotted against the concentration of the SCFA in order to get the calibration graph for each SCFA.

### 2.8. Statistical Analysis

All tests were done at least three times as independent repetitions and all data expressed as means ± standard deviations. Significant differences among strains, fermentation times or treatments were evaluated by one-way analysis of variance (ANOVA) and Tukey tests. The statistical significance level was defined as *p* < 0.05. Data was analyzed using the statistical software Minitab version 17 (IL, USA).

## 3. Results

### 3.1. Bacterial Survial to Simulated Gastric and Intestinal Fluids

Table 2 contains the results of the reduction in the different viable bacteria counts after incubation in simulated gastric or intestinal conditions. Initial culture counts were 7.2 ± 0.6 log (CFU/mL) for all tests. *Lactobacillus acidophilus* La3, *Lactiplantibacillus plantarum* V3, *L. acidphilus* La56, *Lactiplantibacillus plantarum* B112, *Lactiplantibacillus plantarum* 299v and *Lacticaseibacillus casei* B93 were the most resistant bacteria to gastric conditions. A comparison among species indicated that both *L. acidophilus* strains showed the lowest reductions in viability after exposure to acidic gastric conditions (1.05 ± 0.31 and 2.03 ± 0.13 log), whereas the three *L. delbrueckii* strains presented significant higher reductions (6.82 ± 0.28, 6.65 ± 0.35, and 6.35 ± 0.03 log). Notwithstanding, considering the whole data set, results suggest a strain dependent behavior rather than a species dependent. For example, the five strains of *Lactiplantibacillus plantarum* exhibited varied survival rates ranging between 1.75 ± 0.89 and 6.98 ± 0.41 log. On the other hand, none of the strains tested herein manifested a dramatic decrease in viable counts after incubation at intestinal conditions. *Lactiplantibacillus plantarum* B112, *Lacticaseibacillus casei* Lc93, *L. acidophilus* La3, *Lactiplantibacillus plantarum* Lp17 and *L. casei* Lc04 were identified as the most resistant strains to intestinal fluid rich in bile acids (0.05 ± 0.01, 0.07 ± 0.01, 0.10 ± 0.02, 0.10 ± 0.01, 0.12 ± 0.05 log of reduction, respectively). Due to their low reduction after exposing to both gastric and intestinal tract conditions, *L. acidophilus* La3, *Lactiplantibacillus plantarum* V3, *L. acidphilus* La56, *Lactiplantibacillus plantarum* B112, *Lactiplantibacillus plantarum* 299v and *Lacticaseibacillus casei* B93 were selected for further testing for resistance to different antibiotics and adhesion to Caco-2 cells.

### 3.2. Evaluation of the Antibiotic Resistance

The antibiotic susceptibility patterns of *Lactiplantibacillus plantarum* V3, *L. acidophilus* La3, *Lacticaseibacillus casei* B93, *Lactiplantibacillus plantarum* B112, *Lactiplantibacillus plantarum* 299v, *L. acidophilus* 56 and *Staphylococcus aureus* ATCC 25923 (control) depicted in Table 3 indicated that all *Lactobacillus* strains were resistant to Cephalotin, Ampicillin, Enoxacin, Penicillin, Amikacin, Gentamicin and Dicloxacilin because they did not show an inhibition halo (0.0 mm). For this reason, these data were not included in Table 3. *L. acidophilus* 56 was the most antibiotic resistant strain being intermedium for Ceftriaxone and Chloramphenicol and resistant to the rest of the antibiotics. *Lacticaseibacillus casei* B93 was the only bacterium resistant to Ceftriaxone and was also resistant to Netilmicin. However, *Lactiplantibacillus plantarum* 299v was the most antibiotic susceptible strain, resulting intermedium for Erythromycine and Netilmicin and susceptible for Trimethoprim + Sulfamethoxazole, Ceftriaxone and Chloramphenicol. Furthermore, it was the only strain susceptible to Trimethoprim + Sulfamethoxazole. *Lactiplantibacillus plantarum* B112 and *Lactiplantibacillus plantarum* 299v profiles were similar, except for Trimethoprim + Sulfamethoxazole. *Lactiplantibacillus plantarum* V3 and *L. acidophilus* La3 presented the same antibiotic susceptibility profile: sensible for Ceftriaxone; intermedium for Erythromycine, Netilmicin and Chloramphenicol; and resistant for the rest of the antibiotics.

### 3.3. In Vitro Adhesion to Caco-2 Cells

Figure 1 depicts adhesion percentage to Caco-2 cells of the best aforementioned strains that are resisted to both gastric and intestinal test conditions. All tested bacteria adhered to Caco-2 cells in relevant amounts (between 10% and 27%). *Lactiplantibacillus plantarum* V3 and *Salmonella enterica* ATCC 13311 proved the highest adhesion, followed in order by *L. acidophilus* La3, *L. acidophilus* La56, *Lactiplantibacillus plantarum* 299v, *Lacticaseibacillus casei* B93 and *Lactiplantibacillus plantarum* B112. Interestingly, the adhesion of *Lactiplantibacillus plantarum* V3 did not significantly differ with the positive control *Salmonella enterica* ATCC 13311. It is also interesting that all bacteria were at least as adherent as *Lactiplantibacillus plantarum* 299v, even *L. acidophilus* La3 and *Lactiplantibacillus plantarum* V3 were more adherent than *Lactiplantibacillus plantarum* 299v.

### 3.4. Prebiotics Fermentation by Potential Probiotics

*Lactiplantibacillus plantarum* V3 and *L. acidophilus* La3 were chosen based on their performance in vitro screening for probiotic properties. Then, our aim was to select the best synbiotic combination of probiotic bacteria with inulin or starch-based SCF. Figure 2 shows the comparisons of the growth kinetics throughout fermentations performed with culture media supplemented with inulin or soluble corn fiber. In control treatments, dextrose was added in a limited amount to appreciate the death phase, represented as a decline in growth. The ability of inulin and starch-based SCF to support the growth of *Lactiplantibacillus plantarum* V3 and *L. acidophilus* La3 was demonstrated. All prebiotic treatments displayed a diauxic growth during fermentation, a kind of growth often observed when microbes are grown in a medium containing two carbohydrates, in this case, consisting of simple sugars and inulin or starch-based SCF. *Lactiplantibacillus plantarum* V3 and *L. acidophilus* La3 were capable to grow in both soluble fiber sources, whereas the *L. acidophilus* La3 had the highest growth when grown in the starch-based SCF medium. *Lactiplantibacillus plantarum* V3 exhibited similar maximum growth in both soluble fiber sources, but adaptation, which was seen as the duration of the second lag phase, was faster when grown in starch-based SCF. Conversely, *L. acidophilus* La3 presented higher growth and faster adaptation in starch-based SCF than in inulin.

Table 4 contains physicochemical parameters of prebiotic fermentations. The change in pH over fermentation time varied according to treatment, but not when the different bacteria strains were compared. Both controls exhibited a more pronounced decrease in pH and increase in acidity, in contrast to both fiber treatments. Meanwhile, acidity values increased during the first 8 h of fermentation, then remained the same except for the *Lactiplantibacillus plantarum* V3 that grew in the control medium, where acidity gradually increased during the first 12 h. In addition, it was observed that *Lactiplantibacillus plantarum* V3 produced higher acidity in contrast to *L. acidophilus* La3 independently of the type of treatment. The °Brix did not vary greatly regarding fermentation time for each treatment. Both controls gave lower °Brix values (5.1–5.3) compared to the prebiotic treatments (5.8–6.6) which showed similar values.

### 3.5. Short and Medium Chain Fatty Acids Production

Figure 3 depicts acetic and butyric acids concentrations throughout fermentation of dextrose, inulin or starch-based SCF by *Lactiplantibacillus plantarum* V3 and *L. acidophilus* La3. Acetic acid was the major SCFA produced in all fermentations comprising more than 98% of the total amount. In contrast, propionic, isobutyric, isovaleric, valeric, isocaproic, caproic and heptanoic acids were found in trace amounts or not even detected (data not shown), with the exception of the broth fermented for 48 h by *Lactiplantibacillus plantarum* V3; this bacterium generated in the inulin treatment of 0.93 ± 0.79 mMol/L isovaleric, 0.50 ± 0.28 mMol/L valeric, 0.90 ± 0.54 mMol/L isocaproic and 0.57 ± 0.32 mMol/L heptanoic acids. Caproic was the only acid that presented significant increase in concentrations over time (*p* < 0.05). Meanwhile, *L. acidophilus* La3 in same conditions yielded, in mMol/L, 0.42 ± 0.05 propionic, 0.15 ± 0.01 isobutyric, 0.61 ± 0.25 isovaleric, 0.14 ± 0.05 valeric and 0.49 ± 0.03 isocaproic acids. The propionic, isobutyric, valeric and isocaproic acids (*p* < 0.05) increased throughout the fermentation time.

There were no important differences (*p* < 0.05) in the production of acetic or butyric acids among the distinct strains of bacteria when cultured with the same carbon source, except for the starch-based SCF fermented for 48 h with *Lactiplantibacillus plantarum* V3, which did not produce any butyric acid. The combinations that generated more SCFA were inulin with either *Lactiplantibacillus plantarum* V3 or *L. acidophilus* La3 (97.1 ± 14.9 and 85.7 ± 4.9 mMol/L of total SCFA, respectively, at 48 h), whereas the combinations that created the lowest amounts were dextrose with *Lactiplantibacillus plantarum* V3 or *L. acidophilus* La3 (64.4 ± 7.5 and 50.3 ± 22.0 mMol/L, respectively at 48 h). Acetic acid concentration decreased at 48 h in both dextrose fermentations by *Lactiplantibacillus plantarum* V3 or *L. acidophilus* La3 but remained constant when inulin or starch-based SCF were fermented. Therefore, significantly higher values of these SCFA were observed in both fiber treatments compared to dextrose fermented for 48 h (*p* < 0.05). Butyric acid only varied between treatments fermented for 48 h and displayed the highest concentrations in treatments fermented with starch-based SCF or inulin by *L. acidophilus* La3 (0.31 and 0.20 mMol/L, respectively).

## 4. Discussion

The purpose of this study was to identify and propose synbiotic combinations for their further application in effective nutraceutical food products. The process to achieve that objective is as follows: first, various LAB, including commercial starter cultures, were screened as probiotics; then, the best LAB fermented inulin or starch-based SCF in order to detect SCFA production. Results showed important differences in the inulin and starch-based SCF ability to promote growth of two highly functional *Lactobacillus* strains in terms of probiotic characteristics and SCFA production.

Their capacity to survive gastrointestinal tract conditions was variable and strain-dependent. For example, it was noted that log reductions oscillated between 1.76–6.98 log (CFU/mL) in four *Lactiplantibacillus plantarum* strains. LAB effectively fermented sugars into acidic end-products and were capable of growing at low pH, notwithstanding that they are neutrophils. The LAB stress responses are intricate and involved several mechanisms. The spectrum of mechanisms differs among species and it is not always clear, especially regarding enzymatic and scavenging activities [26]. Probably, the resistance to acid and bile salts of any given strain depends on its individual capacity to use several stress response mechanisms and their adaptive evolution. *Lactiplantibacillus plantarum* 299v and *Lacticaseibacillus rhamnosus* GG were included in the present study as reference strains because of their high functionality and multiple benefits to host health [27]. Interestingly, *Lacticaseibacillus rhamnosus* GG had high reductions in viable counts after exposure to gastric conditions. Monteagudo-Mera and collaborators [9] tested this particular strain against simulated gastric and intestinal fluids at various pHs, bile concentrations and incubation times and concluded that they had different survival rates depending on combined effects. The authors reported similar viability reductions at similar conditions. It is remarkable that *L. acidophilus* La3, *Lactiplantibacillus plantarum* V3 and *L. acidophilus* La56 were significantly more acid resistant compared to *Lactiplantibacillus plantarum* 299v. All strains studied showed high survival rates to bile salts. These results are consistent with a previous investigation [28]. These authors found that strains of *L. delbruekii*, *L. lactis*, *L. acidophilus*, *Lacticaseibacillus casei* and *Lacticaseibacillus rhamnosus* were able to grow when exposed to 0.5% of bile salts which is the same concentration used in the current study.

*Lactiplantibacillus plantarum* V3, *L. acidophilus* La3, *Lacticaseibacillus casei* B93, *Lactiplantibacillus plantarum* B112, *Lactiplantibacillus plantarum* 299v and *L. acidophilus* 56 were all resistant to Cephalotin, Ampicillin, Enoxacin, Penicillin, Amikacin, Gentamicin and Dicloxacilin. Their susceptibility when exposed to the rest of antibiotics varied among strains. Earlier reports have stated that lactobacilli had a wide range of antibiotic resistances naturally and that the susceptibility patterns differ greatly among species [29]. Many lactobacilli strains with resistance to Gentamicin, Penicillin and Ampicillin have been reported. Likewise, the resistance to Kanamycin, Vancomycin, Methicillin, Cloxacillin, Nalidixic acid, Aztreonam, Cicloserin, Polymyxin B and Spectinomycin have been commonly found. On the other hand, lactobacilli have been documented to be sensitive to Rifampicin, Bacitracin, Clindamycin and Novobiocin. Finally, *Lactobacillus* susceptibility to Cephalothin, Erythromycin, Chloramphenicol, Gentamicin, Lincomycin, Metronidazole, Neomycin, Streptomycin and Tetracycline was noted to be variable depending on the species [30]. *Lactobacillus* strains with non-transmissible antibiotic resistances usually do not pose a safety concern, for example, several species of lactobacilli are intrinsically resistant to Vancomycin, many of them have a long history of safe use in foods and there is no indication that their Vancomycin resistance could be transferred to other bacteria [31]. The bacteria *L. acidophilus* La56 and *Lacticaseibacillus casei* B93 presented the highest resistance to antibiotics, both being resistant to Netilmicin. However, *L. acidophilus* La56 and *Lacticaseibacillus casei* B93 were the only ones that showed resistance to Erythromycin and Ceftriaxon, respectively. Therefore, these resistant mutants were discarded for further testing. Thus, it is necessary to conduct more research aimed to check the ability of proposed probiotic strains or bacteria starter cultures to act as a donor of conjugative antibiotic resistance genes, in order to identify, above all, if bacterial resistance is the transferable type and to evaluate related food safety issues.

It was shown that all strains examined complied with the relevant parameter of adhesion to the intestinal epithelium. *Salmonella enterica* ATCC 13311 was chosen as a control since this bacteria possesses high capacity to adhere and colonize the intestine and its pathogenicity is largely due to these capabilities. In this context, it is notable that *Lactiplantibacillus plantarum* V3 was able to adhere to cells in a manner comparable to *Salmonella* (23.9 ± 2.3% and 27.0 ± 10.7%, respectively). Most *Lactiplantibacillus plantarum* strains can adhere to cell lines of intestinal origin because they have a mannose binding adhesion site. For example, the strain *Lactiplantibacillus plantarum* 299v readily adapts to establish itself in the human intestinal mucosa given that it has a specific mechanism for adhesion consisting of binding to a mannosylated cell-bound receptor. The competition between *Lactiplantibacillus plantarum* strains and pathogenic bacteria for these receptors is known to reduce the pathogens adherence to intestinal cells, preventing gastrointestinal infections [32]. Moreover, *Lactiplantibacillus plantarum* 299v has been reported as a strain with high adhesion capacity [6], thus it was used as a reference. Taking this finding into consideration, the bacteria employed herein showed relevant capability to adhere to the intestinal epithelium since all of them exhibited similar or better adhesion in comparison to *Lactiplantibacillus plantarum* 299v.

Some of the strains have been previously tested for its probiotic properties, both consistent and contrasting results have been found. *Lactobacillus acidophilus* La3 showed a net reduction of 1.15 Log after gastric and intestinal tests, high survival rates have been previously found, populations were reduced by 2.9 log [33] or even no decrease were observed [34]. A former study [35] evaluated *Lactobacillus* strains isolated from Iranian traditional food products and human feces, including *Lactobacillus acidophilus* ATCC 4356 and *Lactiplantibacillus plantarum* ATCC 14917 as controls, reported adhesion capacity 2.4% and 4.6 in HT-29 cells and 21% and 69% survival rate in gastrointestinal in vitro test, respectively; in contrast, among all studied strains, *Lactobacillus acidophilus* ATCC 4356 showed the lowest percentage of adhesion and survival rate, while in the present study, this strain rated in the best five strains (with 14% and 49%, respectively); however, important methodological differences should be taken into account, for example, the cell line used (Caco-2). *Lactiplantibacillus plantarum* 14917 resistance was evaluated in bile salts, showing 65.8% of growth; also in addition, survival in gastric conditions were monitored at different times while gastric solution was gradually acidified from pH 5.0 to 2.2, resulting in 3.1 log reduction at 60 min [7]; it is interesting that, growth is reported while reductions have been previously described. Commercial lactic acid bacteria were screened for survival in simulated gastric juice and bile; *Lactobacillus delbrueckii* 11842 was included as control and it was found tolerant to gastric and small intestinal transit (55% net survival) [8], opposite to results previously described in results section (2.5% net survival). *Lactococcus lactis* ATCC 11454 was not considered for further study because of its high reduction in the sum of gastric and intestinal in vitro tests (7.02 and 0.36 respectively), however, this strain was evaluated using a gastrointestinal dynamic model, which consisted of four compartments simulating stomach, duodenum, jejunum and ileum interconnected in series by computer controlled peristaltic valve pumps, where lower reductions in gastric conditions and higher reductions in intestinal phases were found (4, 6, 0 and 1 log, respectively) [11]. In [9] *Lacticaseibacillus casei* 393 and *Lactobacillus rhamnosus* 53103 were evaluated at various pH, time and levels and bile salts concentration levels, finding high survival rates at all conditions. However, when they measured adhesion in Caco-2 cells, low adhesion capacity was found in *Lacticaseibacillus* casei 393 comparing with *Lacticaseibacillus rhamnosus* 53103 (4% and 9%, respectively). *Lacticaseibacillus* 53103 has been used as control to evaluate *Lactobacillus* isolated from fermented olives [12], a 3-log reduction was reported at pH 2.5 and 34% of adhesion in Caco-2 cells. Contrasting adhesion percentage between studies is an example of how differences in methodology could impact results.

In summary of the in vitro tests, it was demonstrated that both *Lactiplantibacillus plantarum* V3 and *L. acidophilus* La3 manifested outstanding ability to survive gastric and intestinal conditions and to adhere to layered Caco-2 cells. Furthermore, they had similar antibiotic resistance compared to *Lactobacillus*. It is also notable that these two bacteria exhibited similar or better performance in contrast to the well-known probiotic bacterium *Lactiplantibacillus plantarum* 299v. Consequently, these promising strains were selected for the prebiotic fermentation experiments.

Although pure culture fermentations do not reflect bacterial interactions in the host, this type of assay is useful for screening potential prebiotic fibers and, subsequently, the selection of synbiotic combinations. *Lactiplantibacillus plantarum* V3 and *L. acidophilus* La3, both facultative heterofermentative bacteria, were studied to test their ability to grow in a starch-based SCF or inulin media and to produce short chain fatty acids. An increase in CFU/mL and acidity and a decrease in the pH of the culture reflect the bacteria capacity to grow on different substrates. As previously stated, diauxic growth (or diauxi) was observed after studying fermentation kinetics. In diauxic growth, which results in separate phases, the bacteria use the available sugars in a sequential manner rather than a simultaneous one. Typically, the two growth stages are separated by a phase of arrested growth, also known as a lag-phase. Diauxic growth is often interpreted as an adaptation to maximize population growth in multi-nutrient environments. Results herein demonstrated that both *Lactiplantibacillus plantarum* V3 and *L. acidophilus* La3 grew: first, using sugars that remained in the purified fibers, then it underwent an adaptation phase in order to grow again by consuming inulin or the starch-based SCF. These findings show the *Lactiplantibacillus plantarum* V3 and *L. acidophilus* La3 capability to metabolize these soluble fiber sources; however, it was noticed that both bacteria metabolized faster starch-based SCF compared to inulin. The difference was evidenced by a shorter lag phase during the switch from the use of sugars to fiber sources. Clinical trials have revealed that inulin and xylooligosaccharides induced the *Bifidobacterium* spp. Growth, whereas the starch-based soluble corn fiber and polydextrose stimulated the growth of bacterium types, such as *Bacteroidetes* and *Firmicutes* [36]. The molecular size or degree of polymerization/substitution is known to influence the prebiotic capacity of fibers. It has been reported that the specific structure of a given fiber affects bacterial growth and immunomodulation. For example, various wheat arabino-xylooligosaccharides differing in their structure exerted dissimilar prebiotic and fermentation properties in rats [37]. Starch-based SCF, commercially named PROMITOR™ Soluble Corn Fiber 85, is composed by a mixture of α 1-6, α 1-4 and α 1-2 glucosidic linkages that confer low digestibility. Inulin has a more complex structure because its chains are constituted by a variable number of fructose units linked by β-(2→1)d-fructosyl-fructose bonds. The inulin chains usually end with only one glucose unit linked through an α-d-glucopyranosyl or α-(1→2) bond, as in sucrose [38]. The different structures of the soluble fibers could explain fermentation patterns, the lower molecular weight and less branched chains of starch-based SCF are easier to ferment in contrast to inulin.

Some polysaccharides cannot be digested by humans, getting to the colon intact, where they are fermented easily by different groups of bacteria species, such as *Lactobacillus*, *Bifidobacterium*, *Roseburia*, and *Faecalibacterium*. These dietary compounds yield various types of SCFA during fermentation in the hindgut, primarily acetate, propionate and butyrate. In particular, acetate and propionate have energetic effects for eukaryotic cells, while butyrate is the preferred energy source for colon cells. The last one is considered as a strong anti-tumor compound for colon cells, it regulates cell multiplication pathways and promotes pro-apoptotic pathways. Medium chain fatty acids containing 5 to 8 carbon chains are effective for weight control and for immune modulation. Moreover, the MCFAs have antimicrobial properties and are inhibitors of some pathogenic organisms [39].

The physicochemical changes and SCFA and MCFA production of the bacteriasuggest that the metabolic pathways followed by the medium rich in dextrose and the fibers media were different. The pronounced decrease in pH and the increase in acidity was comparatively higher in dextrose fermentations, which was probably due to the generation of higher lactic acid amounts, assuming that acidity measured is given by lactic acid, which is the principal end product of *Lactobacillus* fermentations.

Inulin fermentations with both *Lactiplantibacillus plantarum* V3 and *L. acidophilus* La3 yielded high quantities of SCFA after 48 h or the advanced phase of fermentation. This effect has been seen in previous studies using *Bifidobacteria* cultures, where the availability of carbon affected the type of fermented compounds and the bacteria growth rate. Conversely, studies on fermentations with carbon limitation indicated the production of acetic and formic acids, whereas fermentations conducted with excess carbohydrates generated high amounts of acetate and lactate. Moreover, when *Clostridium perfringens* was grown under carbohydrate excess conditions, the bacterium increased lactate formation at the expense of acetate [40].

LAB metabolism normally produces lactic acid as the major end-product of the glycolytic fermentation. Regarding SCFAs pathways, *Lactobacillus* can produce them by fermenting carbohydrates end-products, such as pyruvate, which is generated during the glycolytic pathway. Additionally, they are formed by the phosphoketolase route in the heterofermenting conditions [41]. In addition to pyruvate, lactic acid has been described to be used in acetic acid formation and to also create acetyl-CoA, which provides two carbons for the elongation of acetate to butyrate via reverse β-oxidation. SCFA are further chain-elongated to medium-chain carboxylic acids through the reverse β-oxidation pathway. However, there has been very few reports on the production of these carboxylic acids. This is the reason why MCFA were seen only in trace amounts or not detected. It is also interesting that inulin fermentations that yielded MCFA did not show significant quantities of both acetic and butyric acids, a possible reason is that these two compounds are known precursors of MCFA. This could also apply in starch-based SCF fermentation by *L. acidophilus* La3, where butyric acid was formed but acetic acid (butyric acid precursor) did not increase.

Only the inulin fermentations generated significant amounts of propionic acid. The highest butyric acid production was obtained by fermentation with the starch-based SCF. These notable distinctions indicate that the fiber type produces different SCFA and MCFA profiles. It is also important to mention that *Lactiplantibacillus plantarum* V3 was not able to produce butyric acid nor any other SCFA or MCFA when fermenting the starch-based SCF. On the other hand, this particular bacterium strain created almost all SCFA or MCFA when fermenting inulin.

The inulin fermentations were comparatively more effective in promoting total SCFA formation. Inulin fermentation for 48 h with *Lactiplantibacillus plantarum* V3 and *L. acidophilus* La3 reported the highest total SCFA values (97.1 mMol/L and 85.7 mMol/L respectively). In addition, it is notable that their respective controls had a significant reduction at 48 h of fermentation (50.3 mMol/L final versus 93.0 initial, for L. acidophilus La3) (*p* < 0.05), which indicates oxidation. Due to this, it is considered the possibility that the initial compounds have been degraded over time; therefore, quantified amounts are the net result of simultaneous compounds production and degradation.

Additionally, to propose combinations as, not only the total, but the best of SCFA production was considered an important parameter to be taken into account in addition to the growth rates achieved and type of compounds produced. Best growth rates were obtained in inulin + *L. acidophilus* La3 and SCF + *L. acidophilus* La3 treatments. Although SCF + *L. acidophilus* La3 did not generate the highest total SCFA concentrations, it manifested fast adaptation and the highest growth. It is notable, as well, that it presented an important production of butyric acid, which is one of the most functional compounds. On the other hand, the highest treatments in total SCFA owe their increase to small increases in various compounds, such as in valeric and isocaproic acids.

Results herein suggest the synbiotic potential of the distinct probiotic/prebiotic combinations. Considering the results reported in this work, inulin + *L. acidophilus* La3 and starch-based SCF + *L. acidophilus* La3 showed the most promising potential as synbiotic pairs. The development of functional foods is a complex process comprising different phases of design and evaluation, this cycle involves defining the bioactive, the target disease, regulatory framework review, food matrix selection, formulation, process design, among other multiple stages [42]. Several steps are still needed before the release of a functional food, however, this screening for potential synbiotics advance the “definition of bioactive” step and provides information for future design of synbiotics for food application. Once a prototype of functional food is developed, further in vivo experiments are required to confirm prebiotic fibers specificity, the beneficial effect in host and the synergistic effect of inulin + *L. acidophilus* La3 and starch-based SCF + *L. acidophilus* La3.

## 5. Conclusions

*Lactiplantibacillus plantarum* V3, *L. acidophilus* La3, *Lactiplantibacillus plantarum* B112 and *Lactiplantibacillus plantarum* 299v can be considered potential probiotics due to their capacity to meet many of the required characteristics and health benefits that need to be further studied in in vivo assays and clinical trials. Additionally, further investigation is needed to assess whether the antibiotic resistance manifested by the proposed probiotics is transferable horizontally to other bacteria that does not belong to the progeny. Starch-based SCF allowed probiotics to achieve high growth and to produce different SCFA, therefore starch-based SCF is considered as a potential prebiotic. More exploration of starch-based SCF as a prebiotic in combination with different probiotics may be also promising. This study is considered a first step to propose effective synbiotics, results herein are useful for the adequate selection of probiotic/prebiotics combinations and, thus, for the future development of efficient functional foods. Further in vivo experiments are needed to evaluate selective utilization of starch-based SCF and inulin and to demonstrate health benefits in the host.

## Figures and Tables

**Figure 1 foods-11-04020-f001:**
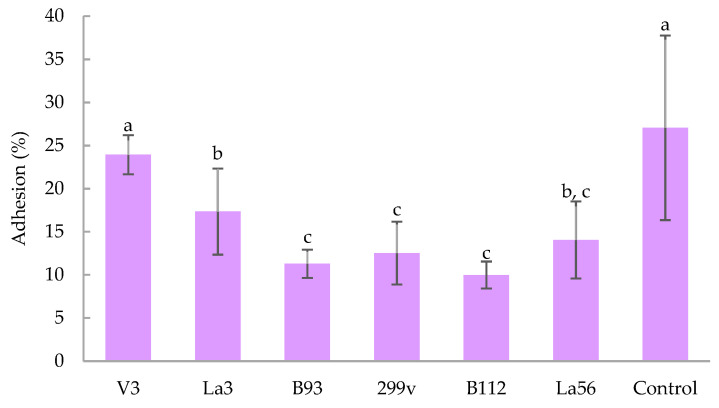
Adhesion of bacteria strains *Lactiplantibacillus plantarum* V3, *L. acidophilus* La3, *Lacticaseibacillus casei* B93, *Lactiplantibacillus plantarum* B112, *Lactiplantibacillus plantarum* 299v, *L. acidophilus* La56 and *Salmonella enterica* ATCC 13311 (Control) to human intestinal epithelial Caco-2 cell cultures. Mean (±standard deviation) of results from three separate experiments. Means with no letter in common are significantly different (*p* < 0.05).

**Figure 2 foods-11-04020-f002:**
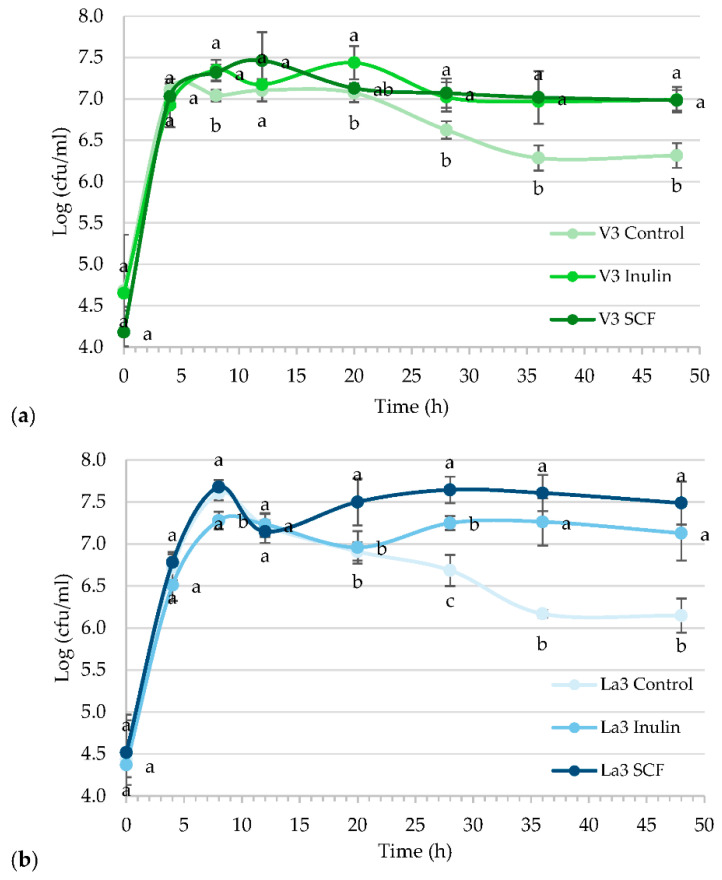
Comparative growth kinetics of *Lactiplantibacillus plantarum* V3 (**a**) and *Lactobacillus acidophilus* La3 (**b**) cultured with dextrose (control), inulin or starch-based soluble corn fiber (SCF) Mean (±standard deviation) of results from three separate experiments. Treatment means with no letter in common within the same time are significantly different (*p* < 0.05).

**Figure 3 foods-11-04020-f003:**
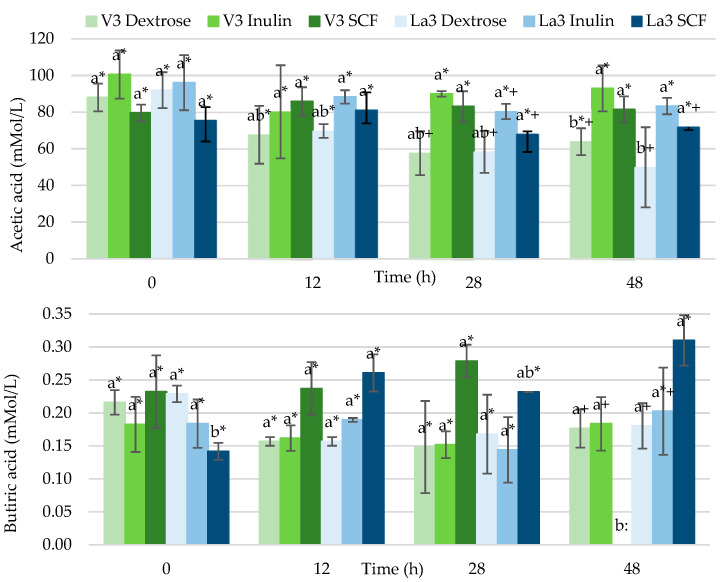
Production of acetic and butyric acids by *Lactiplantibacillus plantarum* V3 and *Lactobacillus acidophilus* La3 throughout 48 h fermentation of dextrose, inulin or starch-based soluble corn fiber (SCF). Acetic and butyric acids concentrations expressed as mMol/L. Mean (±standard deviation) of results from three separate experiments. Bars with no letter in common for each fermentation time are significantly different and bars with no symbol in common for each fermentation treatment are significantly different (*p* < 0.05).

**Table 1 foods-11-04020-t001:** Sources and codification of the different microorganisms used to assess probiotic properties.

Species	Collection *	Strain	Code
*Lactiplantibacillus plantarum*	DSM	9843 (299v)	299v
*Lactiplantibacillus plantarum*	ATCC	14917	Lp 17
*Lactiplantibacillus plantarum*	Sacco (LMG)	P-20353	B112
*Lactiplantibacillus plantarum*	Sacco	V3	V3
*Lacticaseibacillus rhamnosus*	ATCC	53103 (GG)	GG
*Lacticaseibacillus rhamnosus*	Sacco (DSM)	21690	SP1
*Lactobacillus delbrueckii*	ATCC	11842	Ld42
*Lactobacillus delbrueckii*	Sacco (DSM)	21471	SP5
*Lactobacillus delbrueckii*	ATCC	9649	Ld49
*Lacticaseibacillus casei*	ATCC	393	Lc93
*Lacticaseibacillus casei*	LMG	6904	Lc04
*Lacticaseibacillus casei*	Sacco (LMG)	P-20352	B93
*Leuconostoc mesenteroides*	ATCC	10830a	Lm0a
*Lactobacillus acidophilus*	Sacco (DSM)	17742	La3
*Lactobacillus acidophilus*	ATCC	4356	La56
*Lactococcus lactis*	ATCC	11454	Ll54

* DSM: German Collection of Microorganisms and Cell Cultures GmbH. ATCC: American Type Culture Collections. LMG: Bacteria Collection of the Laboratory of Microbiology, University of Ghent. Sacco: commercial lactic cultures (SACCO SRL, Cadorago CO, Italy) including collection culture deposit number, if any.

**Table 2 foods-11-04020-t002:** Effect of simulated gastric or intestinal conditions on count reduction as affected by the type of bacteria.

	Logaritmic (cfu/mL) Reduction	
Bacteria	Gastric	Intestinal
*Lactobacillus acidophilus* La3	1.05 ± 0.31 ^i^	0.10 ± 0.02 ^f^
*Lactobacillus acidophilus* La56	2.03 ± 0.13 ^h^	1.61 ± 0.14 ^a^
*Lacticaseibacillus casei* B93	5.11 ± 0.45 ^e,f^	0.81 ± 0.17 ^b^
*Lacticaseibacillus casei* Lc93	7.63 ± 0.08 ^a^	0.07 ± 0.01 ^f^
*Lacticaseibacillus casei* Lc04	5.47 ± 0.39 ^e^	0.12 ± 0.05 ^f^
*Lactobacillus delbrueckii* SP5	6.82 ± 0.28 ^b,c^	0.16 ± 0.02 ^e,f^
*Lactobacillus delbrueckii* Ld42	6.65 ± 0.40 ^b,c,d^	0.37 ± 0.09 ^c,d,e^
*Lactobacillus delbrueckii* Ld49	6.35 ± 0.03 ^c,d^	1.00 ± 0.04 ^b^
*Lactiplantibacillus plantarum* V3	1.76 ± 0.89 ^h,i^	0.21 ± 0.06 ^d,e,f^
*Lactiplantibacillus plantarum* B112	4.10 ± 0.40 ^g^	0.05 ± 0.01 ^f^
*Lactiplantibacillus plantarum* 299v	4.50 ± 0.35 ^f,g^	0.18 ± 0.04 ^e,f^
*Lactiplantibacillus plantarum* Lp17	6.98 ± 0.41 ^a,b,c^	0.10 ± 0.01 ^f^
*Lacticaseibacillus rhamnosus* SP1	6.98 ± 0.30 ^a,b,c^	0.52 ± 0.15 ^c^
*Lacticaseibacillus rhamnosus* GG	7.33 ± 0.02 ^a,b^	0.25 ± 0.04 ^d,e,f^
*Lactococcus lactis* Ll54	7.02 ± 0.18 ^a,b,c^	0.36 ± 0.03 ^c,d,e^
*Leuconostoc mesenteroides* Lm0a	7.49 ± 0.27 ^a,b^	0.17 ± 0.01 ^e,f^

Mean (±standard deviation) of results from three separate experiments. Means with no letter in common within the same column are significantly different (*p* < 0.05).

**Table 3 foods-11-04020-t003:** Antibiotic susceptibility profile of bacterial strains *Lactiplantibacillus plantarum* V3, *L. acidophilus* La3, *Lacticaseibacillus casei* B93, *Lactiplantibacillus plantarum* B112, *Lactiplantibacillus plantarum* 299v, *L. acidophilus* 56 and *Staphylococcus aureus* ATCC 25923 (control).

	Diameter of the Inhibition Halo (mm)				
Strain	E	SXT	CRO	NET	C
V3	17.3 ± 0.2 I	0.0 ± 0.0 R	28.9 ± 0.2 S	12.6 ± 0.1 I	19.1 ± 0.4 I
La3	15.4 ± 0.3 I	0.0 ± 0.0 R	27.7 ± 0.3 S	11.9 ± 0.2 I	17.6 ± 0.2 I
B93	24.0 ± 0.5 S	0.0 ± 0.0 R	0.0 ± 0.0 R	10.7 ± 0.2 R	26.7 ± 0.1 S
B112	17.7 ± 0.4 I	0.0 ± 0.0 R	36.1 ± 0.1 S	11.2 ± 0.3 I	27.3 ± 0.1 S
299v	14.9 ± 0.1 I	24.6 ± 0.3 S	31.8 ± 0.0 S	11.2 ± 0.3 I	25.6 ± 0.1 S
La56	0.0 ± 0.0 R	0.0 ± 0.0 R	18.2 ± 0.0 I	8.3 ± 0.4 R	9.1 ± 0.0 I
*S. aureus*	19.7 ± 0.1 I	28.7 ± 0.1 S	16.6 ± 1.2 I	22.8 ± 0.4 S	22.8 ± 0.2 S

Mean (±standard deviation) of results from three separate experiments. Interpretation: S, sensible; I, intermedium; R, resistant; according to standard break point tables. E: erythromycine (15 µg); SXT: trimethoprim + sulfamethoxazole (25 µg); CRO: ceftriaxone (30 µg); NET: netilmicin (30 µg); C: chloramphenicol (30 µg).

**Table 4 foods-11-04020-t004:** Changes in pH, acidity and °Brix throughout 48 h fermentation of dextrose, inulin or starch-based soluble corn fiber (SCF) with *Lactiplantibacillus plantarum* V3 or *Lactobacillus acidophilus* La3.

Bacteria/Analysis	Carbon Source	Time (h)							
		0	4	8	12	20	28	36	48
*Lactiplantibacillus plantarum* V3									
pH	Dextrose	5.31 ± 0.03 ^a^	5.22 ± 0.09 ^a^	4.35 ± 0.20 ^b^	3.92 ± 0.01 ^c^	3.92 ± 0.06 ^c^	3.94 ± 0.01 ^c^	3.97 ± 0.03 ^c^	3.93 ± 0.03 ^b^
	Inulin	5.36 ± 0.05 ^a^	5.26 ± 0.08 ^a^	4.68 ± 0.07 ^a^	4.62 ± 0.07 ^a^	4.59 ± 0.09 ^a^	4.60 ± 0.05 ^a^	4.62 ± 0.05 ^a^	4.59 ± 0.08 ^a^
	SCF	5.34 ± 0.04 ^a^	5.22 ± 0.08 ^a^	4.66 ± 0.14 ^a^	4.50 ± 0.06 ^b^	4.49 ± 0.06 ^b^	4.52 ± 0.03 ^b^	4.55 ± 0.03 ^b^	4.52 ± 0.06 ^a^
Acidity	Dextrose	0.57 ± 0.05 ^a^	0.59 ± 0.05 ^ab^	1.08 ± 0.06 ^a^	1.31 ± 0.08 ^a^	1.34 ± 0.07 ^a^	1.34 ± 0.07 ^a^	1.35 ± 0.06 ^a^	1.40 ± 0.08 ^a^
	Inulin	0.54 ± 0.00 ^ab^	0.60 ± 0.07 ^a^	0.87 ± 0.05 ^ab^	0.86 ± 0.05 ^b^	0.86 ± 0.05 ^b^	0.86 ± 0.08 ^b^	0.87 ± 0.05 ^b^	0.89 ± 0.04 ^b^
	SCF	0.41 ± 0.20 ^abc^	0.47 ± 0.20 ^abc^	0.66 ± 0.31 ^bc^	0.65 ± 0.30 ^bc^	0.64 ± 0.29 ^bc^	0.67 ± 0.30 ^bc^	0.65 ± 0.30 ^bc^	0.66 ± 0.31 ^bc^
°Brix	Dextrose	5.09 ± 0.13 ^a^	5.33 ± 0.32 ^b^	5.27 ± 0.12 ^b^	5.14 ± 0.07 ^b^	5.42 ± 0.41 ^b^	5.17 ± 0.11 ^b^	5.27 ± 0.12 ^b^	5.08 ± 0.15 ^b^
	Inulin	6.11 ± 0.31 ^a^	6.34 ± 0.44 ^a^	6.46 ± 0.19 ^a^	6.42 ± 0.12 ^a^	6.43 ± 0.16 ^a^	6.39 ± 0.20 ^a^	6.51 ± 0.13 ^a^	6.37 ± 0.20 ^a^
	SCF	6.07 ± 0.31 ^a^	6.29 ± 0.48 ^a^	6.41 ± 0.11 ^a^	6.36 ± 0.17 ^a^	6.31 ± 0.19 ^a^	6.36 ± 0.21 ^a^	6.39 ± 0.09 ^a^	6.28 ± 0.22 ^a^
*Lactobacillus acidophilus* La3									
pH	Dextrose	5.35 ± 0.04 ^a^	5.22 ± 0.07 ^a^	4.38 ± 0.20 ^b^	3.92 ± 0.02 ^c^	3.91 ± 0.05 ^c^	3.93 ± 0.03 ^c^	3.93 ± 0.02 ^c^	3.93 ± 0.02 ^b^
	Inulin	5.37 ± 0.05 ^a^	5.25 ± 0.07 ^a^	4.69 ± 0.10 ^a^	4.62 ± 0.07 ^a^	4.59 ± 0.08 ^a^	4.61 ± 0.05 ^a^	4.59 ± 0.07 ^ab^	4.60 ± 0.08 ^a^
	SCF	5.34 ± 0.05 ^a^	5.23 ± 0.08 ^a^	4.66 ± 0.13 ^a^	4.49 ± 0.05 ^b^	4.49 ± 0.06 ^b^	4.52 ± 0.03 ^b^	4.53 ± 0.03 ^b^	4.52 ± 0.05 ^a^
Acidity	Dextrose	0.31 ± 0.18 ^bc^	0.35 ± 0.18 ^bcd^	0.50 ± 0.28 ^cd^	0.48 ± 0.27 ^cd^	0.48 ± 0.27 ^cd^	0.51 ± 0.29 ^bcd^	0.50 ± 0.28 ^cd^	0.51 ± 0.29 ^cd^
	Inulin	0.22 ± 0.14 ^c^	0.25 ± 0.15 ^cd^	0.36 ± 0.22 ^cd^	0.35 ± 0.22 ^cd^	0.35 ± 0.22 ^cd^	0.36 ± 0.21 ^cd^	0.36 ± 0.22 ^cd^	0.36 ± 0.23 ^cd^
	SCF	0.17 ± 0.10 ^c^	0.20 ± 0.11 ^d^	0.28 ± 0.16 ^d^	0.27 ± 0.15 ^d^	0.27 ± 0.15 ^d^	0.29 ± 0.16 ^d^	0.28 ± 0.16 ^d^	0.29 ± 0.16 ^d^
°Brix	Dextrose	5.22 ± 0.52 ^a^	5.49 ± 0.32 ^b^	5.42 ± 0.08 ^b^	5.22 ± 0.12 ^b^	5.22 ± 0.04 ^b^	5.20 ± 0.12 ^b^	5.29 ± 0.11 ^b^	5.19 ± 0.18 ^b^
	Inulin	5.83 ± 0.46 ^a^	6.31 ± 0.37 ^a^	6.33 ± 0.16 ^a^	6.33 ± 0.13 ^a^	6.64 ± 0.37 ^a^	6.32 ± 0.19 ^a^	6.34 ± 0.12 ^a^	6.27 ± 0.23 ^a^
	SCF	6.09 ± 0.39 ^a^	6.59 ± 0.18 ^a^	6.42 ± 0.18 ^a^	6.39 ± 0.16 ^a^	6.30 ± 0.21 ^a^	6.34 ± 0.24 ^a^	6.36 ± 0.19 ^a^	6.31 ± 0.21 ^a^

Mean (±standard deviation) of results from three separate experiments. Columns with no letter in common for each fermentation treatment are significantly different (*p* < 0.05).

## Data Availability

The data are available from the corresponding author.

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
