# Peer review of "Assessment of Potential Probiotic and Synbiotic Properties of Lactic Acid Bacteria Grown In Vitro with Starch-Based Soluble Corn Fiber or Inulin"

_foods, 2022, doi:10.3390/foods11244020_

Round 1

Reviewer 1 Report

The work entitled “Assessment of potential probiotic bacteria and synbiotic capacities of lactic acid bacteria grown in vitro with starch-based soluble corn fiber or inulin” characterized different commercial strains of LAB in terms of their potential probiotic and synbiotic properties, the latter when combined with inulin and starch-based soluble corn fiber. In general, the work is well written and organized. The results were shown clearly with graphs and tables and properly discussed. Nevertheless, I think the overall quality of this work could improve if the authors consider the following suggestions:

Title: I find it confusing, maybe it could be reformulated as follows: “Assessment of potential probiotic and synbiotic properties of lactic acid bacteria grown in vitro with starch-based soluble corn fiber or inulin”

Introduction: the authors could use a more updated definition of probiotics:

-Hill C, Guarner F, Reid G, Gibson GR, Merenstein DJ, Pot B, Morelli L, Canani RB, Flint HJ, Salminen S, Calder PC, Sanders ME. Expert consensus document. The International Scientific Association for Probiotics and Prebiotics consensus statement on the scope and appropriate use of the term probiotic. Nat Rev Gastroenterol Hepatol. 2014 Aug;11(8):506-14. doi: 10.1038/nrgastro.2014.66. Epub 2014 Jun 10. PMID: 24912386.

Introduction: the authors should mention if probiotic properties were previously assessed for the strains selected for this study.

Line 31: I would rather say “strains” instead of “species”, as the probiotic properties are strain-dependent.

Line 55: Include this reference to define prebiotics:

-Gibson, G., Hutkins, R., Sanders, M. et al. Expert consensus document: The International Scientific Association for Probiotics and Prebiotics (ISAPP) consensus statement on the definition and scope of prebiotics. Nat Rev Gastroenterol Hepatol 14, 491–502 (2017). https://doi.org/10.1038/nrgastro.2017.75.

Line 79: Separate “been” from “the”.

Lines 92-94: Could the authors confirm this? is there a reference they can mention?

The authors should use the new nomenclature for lactobacilli throughout the whole manuscript:

-Zheng J., Wittouck S., Salvetti E. et al., (2020). A taxonomic note on the genus Lactobacillus: Description of 23 novel genera, emended description of the genus Lactobacillus Beijerink 1901, and union of Lactobacillaceae and Leuconostocaceae. https://doi.org/10.1099/ijsem.0.004107 ; open access DOI: https://doi.org/10.7939/r3-egnz-m294.

Lines 109, 150, 186, 211: the authors should express the speed in “x g” (g in italics) instead of “rpm”. This must be checked in the whole manuscript.

Line 173: I did not find this part clear enough. What was the final sugar concentration of all the treatments? MRS contains 2%. Why did the authors choose 2.4 g/L? did the control have the same final concentration? This should be clarified as all the treatments need to have the same sugar concentration.

Line 180: why was the treatment with alpha-amylase done here when it was not performed for the strains (section 2.3)?

Line 187: This sterilization step might have affected the final structure of carbohydrates in the media. I think a less aggressive heat treatment would have been more appropriate. Can the authors explain this?

Lines 195-197: Could the authors provide more details about this technique? Or include a reference?

Line 201: SYB or SYBR green?

Line 245: The table should express the results in the same way, showing the first significant digit of the error.

Line 248: Replace exhibit by exhibited.

Lines 270, 274, 287: “lactobacillus” is the genus and should be correctly written.

Table 3: could the authors define the criteria used for S, I and R?

Figure 2: Error bars are missing.

Table 4: did not the authors perform statistical analysis here?

Discussion: more discussion about former studies of these strains should be mentioned, regarding both their technological and probiotic properties. For instance, I found the following works (but the authors need to consider carefully and mention previous characterization of all the strains used):

https://doi.org/10.1016/j.lwt.2016.10.025

https://link.springer.com/article/10.1007/s12602-017-9346-y

https://doi.org/10.1016/j.foodres.2018.08.005

https://doi.org/10.1111/ijfs.15927

https://doi.org/10.1016/j.idairyj.2021.105237

https://doi.org/10.1016/j.lwt.2021.111083

https://doi.org/10.1016/j.lwt.2016.08.009

https://doi.org/10.1111/ijfs.15927

Line 386: The word “stains” should not be in italics.

Line 444: L. plantarum should be in italics.

Line 498: I am not sure about this hypothesis as the levels of lactic acid were not determined.

Lines 527-531: I think this comparison is not adequate because an in vivo model is completely different from any in vitro model, specially when talking about sugar fermentation as the composition of the intestinal microbiota is more complex.

Line 566: I do not think these strains can be considered as probiotics, but they could be potential probiotics whose health benefits need to be further studied in in vivo assays and clinical trials.

Author Response

Thanks for their comments. These comments are very valuable to improve the article. The answers to each of the points are presented below.

Reviewer 1

The work entitled “Assessment of potential probiotic bacteria and synbiotic capacities of lactic acid bacteria grown in vitro with starch-based soluble corn fiber or inulin” characterized different commercial strains of LAB in terms of their potential probiotic and synbiotic properties, the latter when combined with inulin and starch-based soluble corn fiber. In general, the work is well written and organized. The results were shown clearly with graphs and tables and properly discussed. Nevertheless, I think the overall quality of this work could improve if the authors consider the following suggestions:

  1. Title: I find it confusing, maybe it could be reformulated as follows: “Assessment of potential probiotic and synbiotic properties of lactic acid bacteria grown in vitrowith starch-based soluble corn fiber or inulin”

As suggested, we change the title. We consider the reviewer option clear and appropriate.

  1. Introduction: the authors could use a more updated definition of probiotics:

Hill C, Guarner F, Reid G, Gibson GR, Merenstein DJ, Pot B, Morelli L, Canani RB, Flint HJ, Salminen S, Calder PC, Sanders ME. Expert consensus document. The International Scientific Association for Probiotics and Prebiotics consensus statement on the scope and appropriate use of the term probiotic. Nat Rev Gastroenterol Hepatol. 2014 Aug;11(8):506-14. doi: 10.1038/nrgastro.2014.66. Epub 2014 Jun 10. PMID: 24912386.

We updated the definition and included the more recent and suggested reference. 

  1. Introduction: the authors should mention if probiotic properties were previously assessed for the strains selected for this study.

We added a pharaghraph regarding previous evaluations of the strains used in this study.

  1. Line 31: I would rather say “strains” instead of “species”, as the probiotic properties are strain-dependent.

Correction done

  1. Line 55: Include this reference to define prebiotics:

Gibson, G., Hutkins, R., Sanders, M. et al. Expert consensus document: The International Scientific Association for Probiotics and Prebiotics (ISAPP) consensus statement on the definition and scope of prebiotics. Nat Rev Gastroenterol Hepatol 14, 491–502 (2017). https://doi.org/10.1038/nrgastro.2017.75.

We changed the definition and included the suggested reference. 

  1. Line 79: Separate “been” from “the”.

Correction done.

  1. Lines 92-94: Could the authors confirm this? is there a reference they can mention?

Reference included.

  1. The authors should use the new nomenclature for lactobacilli throughout the whole manuscript:

Zheng J., Wittouck S., Salvetti E. et al., (2020). A taxonomic note on the genus Lactobacillus: Description of 23 novel genera, emended description of the genus Lactobacillus Beijerink 1901, and union of Lactobacillaceae and Leuconostocaceae. https://doi.org/10.1099/ijsem.0.004107 ; open access DOI: https://doi.org/10.7939/r3-egnz-m294.

We change the genera Lactobacillus plantarum and L. rhamnosus for Lactiplantibacillus plantarum and Lacticaseibacillus rhamnosus respectively, acording to the new nomenclature. 

  1. Lines 109, 150, 186, 211: the authors should express the speed in “x g” (g in italics) instead of “rpm”. This must be checked in the whole manuscript.

Speed was converted to g force and changed

  1. Line 173: I did not find this part clear enough. What was the final sugar concentration of all the treatments? MRS contains 2%. Why did the authors choose 2.4 g/L? did the control have the same final concentration? This should be clarified as all the treatments need to have the same sugar concentration.

The inulin and starch-based SCF purities were 88 and 87.5% and contained sugar concentrations of 12 and 2% respectively, because different purities and sugar content, when we were doing calculus, it was easier for us to match all treatments to 2.4% of sugars instead of 2%, even if it was a slightly higher concentration than commercial MRS. On the other hand, total fiber content in inulin and starch-based SCF is higher than control because of the fibers, and both fiber treatments were calculated to be almost equal in fiber (17.6% and 17.5% considering slightly different purities) and equal sugar content. Reading those lines again we agreed the redaction is not clear, because it seems like control has less sugar instead of fiber, so we changed the redaction.

  1. Line 180: why was the treatment with alpha-amylase done here when it was not performed for the strains (section 2.3)?

It is because these sections have different objectives. In section 2.3 we aimed to test gastric and intestinal survival of bacteria, considering that there are important criteria to considering them probiotics. In section 2.6, taking into consideration that prebiotics are fermented by probiotics after passing through the gastrointestinal tract, all MRS broths were subjected to simulated complete digestion, which includes alpha-amylase treatment; in this case, bacteria were added after in vitro digestion of prebiotics in a known amount to run the kinetics.

  1. Line 187: This sterilization step might have affected the final structure of carbohydrates in the media. I think a less aggressive heat treatment would have been more appropriate. Can the authors explain this?

Sterilization step weas included because we must be sure that the inoculated probiotic was the only microorganism performing the fermentation, also, this process inactivates enzymes used in the digestion. Reactions we could expect in carbohydrates derived from sterilization are Maillard reaction in sugars and starch-based SCF retrogradation. Maillard reaction is evidenced by color changes, usually, MRS broth becomes browner after sterilization. No more darkening than usual was appreciated in MRS treatments. On the other hand, retrogradation may affect carbohydrate utilization, however, bacteria were able to carry on the fermentation.

  1. Lines 195-197: Could the authors provide more details about this technique? Or include a reference?

A reference with detailed technique explanation was added.

  1. Line 201: SYB or SYBR green?

A letter was missing, we correct SYB for SYBR.

  1. Line 245: The table should express the results in the same way, showing the first significant digit of the error.

As requested, table 2 results was modified to show the first significant digit of the error.

  1. Line 248: Replace exhibit by exhibited.

Correction done.

  1. Lines 270, 274, 287: “lactobacillus” is the genus and should be correctly written.

Taking into consideretion that those pharagraphs are comparing many strains, included Lactiplantibacillus plantarum which is no more a Lactobacillus, we changed or deleted the word lactobacillus.

  1. Table 3: could the authors define the criteria used for S, I and R?

In section 2.4 we added some details to clarify and we included this information in the footnote of the table 3.

  1. Figure 2: Error bars are missing.

We didnt include the error bars because this créate an exces of saturation of lines in the figure. As alternative, to include error bars, we splited the figure in two. You could notice a different format, this is because our software licences expired and we used Excel intead. 

  1. Table 4: did not the authors perform statistical analysis here?

Statical analysis was included.

  1. Discussion: more discussion about former studies of these strains should be mentioned, regarding both their technological and probiotic properties. For instance, I found the following works (but the authors need to consider carefully and mention previous characterization of all the strains used):

https://doi.org/10.1016/j.lwt.2016.10.025

https://link.springer.com/article/10.1007/s12602-017-9346-y

https://doi.org/10.1016/j.foodres.2018.08.005

https://doi.org/10.1111/ijfs.15927

https://doi.org/10.1016/j.idairyj.2021.105237

https://doi.org/10.1016/j.lwt.2021.111083

https://doi.org/10.1016/j.lwt.2016.08.009

https://doi.org/10.1111/ijfs.15927

We searched for more and recent former studies regarding the properties of used probiotic strains and they were disused, and the reference included.

Recommended literature above focus in Lactobacillus acidophilus LA3 survival evaluation protected encapsulated or in a food matrix, Technological properties and survival in a food matrix is covered in the later study yet to be published, in it, we fermented germinated maize with Lactobacillus acidophilus LA3 to obtain a beverage similar to traditional prehispanic beverages in Mexico. Fermentation parameters were optimized, and nutritional and technological properties were evaluated. Then, we consider the recommended bibliography more appropriate to be included in the next publication. However, when therecomended study included a non-protected control, we could use it to compare with our results, in that case the reference was discussed.

  1. Line 386: The word “stains” should not be in italics.

Correction done

  1. Line 444: L. plantarum should be in italics.

Correction done

  1. Line 498: I am not sure about this hypothesis as the levels of lactic acid were not determined.

We mesured acidity and expressed it as lactic acid equivalents, considering that lactic acid is mein product of fermentation by Lactobacillus. However, it is true your afirmation, we did not determine especifically lactic acid, so, we rewritten to mention that this is a posible explanation.

  1. Lines 527-531: I think this comparison is not adequate because an in vivo model is completely different from any in vitro model, specially when talking about sugar fermentation as the composition of the intestinal microbiota is more complex.

Comparison in lines 527-531 was eliminated, considering that this statement regarding the differences between in vitro and in vivo models is correct.

  1. Line 566: I do not think these strains can be considered as probiotics, but they could be potential probiotics whose health benefits need to be further studied in in vivo assays and clinical trials.

We rewriten this sentence and include the aseveration indicated.

Reviewer 2 Report

The manuscript entitled: "Assessment of potential probiotic bacteria and synbiotic capacities of lactic acid bacteria grown in vitro with starch-based soluble corn fiber or inulin" is in the field of food microbiology and biotechnology. The research design is acceptable and the objectives of the research are aligned with the journal's aims and scopes. There are some comments that need to address by the authors as follows:

1- Title: Revise the title; a good research title should shows the "Why" and "How" of the research.

2- Abstract: Improve the abstract by supporting the results with some quantitative data.

3- Keywords: Choose keywords other than the main words of the title. It will improve the visibility of the article.

4- Introduction: Try to use the most recent literature in the field.

5- Subheadings: I recommend checking and improving all subheadings; For example: "2.4. Sensitivity to antibiotics" modify to "2.4. Evaluation of the antibiotic resistance" or maybe a better informative subheading.

6- All methods should cite a proper reference(s) either from a published article or a standard method.

7- In the notes below tables or figures you need to mention data represents for example mean (number of replication) ± SD (or may SE).

8- Improve the quality of the figures by drawing solid filled colours instead of using patterns. 

9- In figures 1 and 3 better to put the significant letters on top of the bars.

10- The conclusion is too long. Justify your hypothesis and may recommend future research in a short paragraph.

11- References: Replace the out date references if possible by recent publications. You can find lots of related articles in MDPI.

Author Response

Thanks for their comments. These comments are very valuable to improve the article. The answers to each of the points are presented below.

Reviewer 2

The manuscript entitled: "Assessment of potential probiotic bacteria and synbiotic capacities of lactic acid bacteria grown in vitro with starch-based soluble corn fiber or inulin" is in the field of food microbiology and biotechnology. The research design is acceptable and the objectives of the research are aligned with the journal's aims and scopes. There are some comments that need to address by the authors as follows:

  1. Title: Revise the title; a good research title should shows the "Why" and "How" of the research.

The title was changed to a clearer one.

  1. Abstract: Improve the abstract by supporting the results with some quantitative data.

The abstract was re writen to include quantitative data.

  1. Keywords: Choose keywords other than the main words of the title. It will improve the visibility of the article.

We modify and added key words.

  1. Introduction: Try to use the most recent literature in the field.

We updated probiotic and prebiotic definitions, and older references were chanced for more recent bibliography.

  1. Subheadings: I recommend checking and improving all subheadings; For example: "2.4. Sensitivity to antibiotics" modify to "2.4. Evaluation of the antibiotic resistance" or maybe a better informative subheading.

Almost all subheadings were change to add details to be more informative.

  1. All methods should cite a proper reference(s) either from a published article or a standard method.

References were included in methods whitout them.

  1. In the notes below tables or figures you need to mention data represents for example mean (number of replication) ± SD (or may SE).

A leyend with that information was added in all tables and figures.

  1. Improve the quality of the figures by drawing solid filled colours instead of using patterns. 

Gray scale in figures was chanched for colors.

  1. In figures 1 and 3 better to put the significant letters on top of the bars.

In figure 1 significant letters were moved to the top. In figure 3, significant letters were placed above, below or by the side of points acordingly, to avoid letters to be overlaped.

  1. The conclusion is too long. Justify your hypothesis and may recommend future research in a short paragraph.

Conclusions section was reducted, results descripctions were deleted and the redaction was changed to focus in the hipótesis justification and future research, as reviewer 2 and 3 recomended.

  1. References: Replace the out date references if possible by recent publications. You can find lots of related articles in MDPI.

We updated probiotic and prebiotic definitions, and older references were chanced for more recent bibliography.

Reviewer 3 Report

Dear Editors and authors, 

1-The abstract of the manuscript needs to add some of the results obtained during the study.

2- The introduction needs to be supported by some recent references. I suggest adding (Zendeboodi, F., Khorshidian, N., Mortazavian, A. M., & da Cruz, A. G. (2020). Probiotic: conceptualization from a new approach. Current Opinion in Food Science, 32, 103-123.,,,,, Li, Y., Wu, Y., Wu, L., Qin, L., & Liu, T. (2022). The effects of probiotic administration on patients with prediabetes: a meta-analysis and systematic review. Journal of Translational Medicine, 20(1), 1-13.,,,,,,,,,,,,,

Al-Sahlany, S. T., & Niamah, A. K. (2022). Bacterial viability, antioxidant stability, antimutagenicity and sensory properties of onion types fermentation by using probiotic starter during storage. Nutrition & Food Science.)

3-The modern nomenclature of lactic acid bacteria should be used throughout the manuscript and Table 1  and Table 2 should be corrected , such as Lactobacillus plantarum,  write Lactiplantibacillus plantarum, And so are the rest of the other species.

4-Page 4 line 129, How many bacteria numbers are there in  the 0.5 McFarland?

5- How many Staphylococcus aureus ATCC 25923 numbers and volume used in Sensitivity to antibiotics method?

6-Some work methods need to add scientific references, such as: Prebiotic fermentation method and Short and medium chain fatty acids formation method.

7-Figure 1 and Figure 3, Small letters should be placed above the columns to be clear.

8-Figure 2 is unclear.

9-The conclusions written are results. Results may not be mentioned in the conclusions chapter. The conclusions chapter must be rewritten.

Author Response

Thanks for their comments. These comments are very valuable to improve the article. The answers to each of the points are presented below.

Reviewer 3

  1. The abstract of the manuscript needs to add some of the results obtained during the study.

The abstract was re writen to include more results in quantitative data.

  1. The introduction needs to be supported by some recent references. I suggest adding (Zendeboodi, F., Khorshidian, N., Mortazavian, A. M., & da Cruz, A. G. (2020). Probiotic: conceptualization from a new approach. Current Opinion in Food Science, 32, 103-123.,,,,, Li, Y., Wu, Y., Wu, L., Qin, L., & Liu, T. (2022). The effects of probiotic administration on patients with prediabetes: a meta-analysis and systematic review. Journal of Translational Medicine, 20(1), 1-13.,,,,,,,,,,,,,

Al-Sahlany, S. T., & Niamah, A. K. (2022). Bacterial viability, antioxidant stability, antimutagenicity and sensory properties of onion types fermentation by using probiotic starter during storage. Nutrition & Food Science.‏)

We updated probiotic and prebiotic definitions, and older references were chanced for more recent bibliography.

  1. The modern nomenclature of lactic acid bacteria should be used throughout the manuscript and Table 1  and Table 2 should be corrected , such as Lactobacillus plantarum,  write Lactiplantibacillus plantarum, And so are the rest of the other species.

We change the genera Lactobacillus plantarum and L. rhamnosus for Lactiplantibacillus plantarum and Lacticaseibacillus rhamnosus respectively, acording to the new nomenclature. 

  1. Page 4 line 129, How many bacteria numbers are there in the 0.5 McFarland?

The 0.5 McFarland standard represents 1.5 x 108 cfu/ml. This data was included.

  1. How many Staphylococcus aureus ATCC 25923 numbers and volume used in Sensitivity to antibiotics method?

Staphylococcus aureus was streak plated using a 0.5 McFarland solution, just like the other strains. We modify the section redaction to clarify this point.

  1. Some work methods need to add scientific references, such as: Prebiotic fermentation method and Short and medium chain fatty acids formation method.

References were included in methods whitout them.

  1. Figure 1 and Figure 3, Small letters should be placed above the columns to be clear.

In figure 1 significant letters were moved to the top. In figure 3, significant letters were placed above, below or by the side of points acordingly, to avoid letters to be overlaped.

  1. Figure 2 is unclear.

We splited the figure in two to improve clarity and we added error bars.

  1. The conclusions written are results. Results may not be mentioned in the conclusions chapter. The conclusions chapter must be rewritten.

Conclusions section was reducted, results descripctions were deleted and the redaction was changed to focus in the hipótesis justification and future research, as reviewer 2 and 3 recomended.

Round 2

Reviewer 1 Report

The manuscript has considerably increased its overall quality after the authors considered all the suggestions. Nevertheless, there are some minor revisions that need to be addressed before its publication, which are listed below. Additionally, I would like to ask the authors to add the line numbers in their answers so I can find the corrections quicker.

line 64: form or from?

lines 197-198: "Fiber concentrations in treatments were 17.6% and 17.5% 197 for inulin, starch-based SCF" respectively? I think that a comma or period is missing after that.

Please check if the final sugar concentration is 2.4% or 2.4 g/L, because, for instance, MRS broth has 2%, equivalent to 20 g/L. I mean, the difference is quite important (according to the manuscript, the concentration is 2.4 g/L, but the authors’ answer says 2.4%).

Line 214: plantarum should be in italics with no capital letter, please check this throughout the manuscript.

Table 2: SD cannot be expressed as 0.0, at least one significant digit different from zero should be informed. For instance, for the first strain under intestinal conditions: Lactobacillus acidophilus La3, 0.10 +- 0.02. This should be checked throughout the table. To have everything homogenous, the authors could inform the results considering the first 2 digits after the decimal point, for all the strains.

Line 300: replace bacteria by bacterium, in singular.

Fig. 2: please add a) and b), and make the changes accordingly in the figure legend.

Line 439: the genus should be in italics.

Line 494-495: I would remove this part: “as mentioned before, methodological differences could make a difference and difficult comparisons”. It is not clear.

Line 496: “reduction” is said twice.

Lines 497-499: there is no need to include the composition of the buffers here, it is a discussion. The authors should focus on the results from previous works and compare them with theirs.

Line 505: what does “lowest results” mean?

Lines 511-512; 514-515: genus and species should be in italics.

Author Response

  1. line 64: form or from?

It must be “from”. Correction done (still line 64).

  1. lines 197-198: "Fiber concentrations in treatments were 17.6% and 17.5% 197 for inulin, starch-based SCF" respectively? I think that a comma or period is missing after that.

Correction done, now it says: “Fiber concentrations in treatments were 17.6% and 17.5% for inulin and starch-based SCF respectively,” (still lines 197-198).

  1. Please check if the final sugar concentration is 2.4% or 2.4 g/L, because, for instance, MRS broth has 2%, equivalent to 20 g/L. I mean, the difference is quite important (according to the manuscript, the concentration is 2.4 g/L, but the authors’ answer says 2.4%).

The correct one is 2.4%, it must have been expressed in percentage as the rest of the components mentioned in that paragraph. The units were corrected (line 193).

  1. Line 214: plantarum should be in italics with no capital letter, please check this throughout the manuscript.

Correction done (line 214). We checked throughout the manuscript for this kind of mistake.

  1. Table 2: SD cannot be expressed as 0.0, at least one significant digit different from zero should be informed. For instance, for the first strain under intestinal conditions: Lactobacillus acidophilus La3, 0.10 +- 0.02. This should be checked throughout the table. To have everything homogenous, the authors could inform the results considering the first 2 digits after the decimal point, for all the strains.

In round 1, we modified table 2 to show the first significant digit of the error considering the correction cited below:

“Line 245: The table should express the results in the same way, showing the first significant digit of the error.”

However, probably we misunderstood and this comment, and it was intended to indicate we should change data in line 245 to be in the same way as in the table 2. We changed data cited in text from this table (lines 272-282) and we undid changes in table 2, to show 2 digits of SD.

  1. Line 300: replace bacteria by bacterium, in singular.

Correction done (line 301).

  1. 2: please add a) and b) and make the changes accordingly in the figure legend.

Correction done (lines 359-362).

  1. Line 439: the genus should be in italics.

Correction done (line 441).

  1. Line 494-495: I would remove this part: “as mentioned before, methodological differences could make a difference and difficult comparisons”. It is not clear.

This part of the sentence was delated (line 496).

  1. Line 496: “reduction” is said twice.

The second “reduction” was delated.

  1. Lines 497-499: there is no need to include the composition of the buffers here, it is a discussion. The authors should focus on the results from previous works and compare them with theirs.

We included test conditions because at the beginning of that paragraph we hypothesized that methodological differences could explain variation between references results (and our own results) for the same strain. However, since we delated that hypothesis (as advised in point 9 above), we removed buffer details in this paragraph (lines 496-528). Also, this paragraph was modified to emphasized and include more comparisons with our results.

  1. Line 505: what does “lowest results” mean?

It means the lowest survival rate and percentage of adhesion among all studied strains; we corrected this sentence to clarify it (line 504).

  1. Lines 511-512; 514-515: genus and species should be in italics.

Correction done (line 521 and 524).

Reviewer 2 Report

The authors fairly revised and addressed all the comments.

Author Response

The authors fairly revised and addressed all the comments.

Reviewer 3 Report

Dear Editors and authors, 

The authors have not made all the required corrections.

The names of the bacteria in Tables 1 and 2 are not written according to modern nomenclature , like Lactobacillus casei .  

Author Response

The names of the bacteria in Tables 1 and 2 are not written according to modern nomenclature, like Lactobacillus casei.  

We are really sorry we didn’t notice all bacteria name changes, now, we carefully checked every used species one by one. In the next table you will find name changes done according to modern taxonomy:

Former name

New name

Correction done in revision Round:

Lactobacillus plantarum

Lactiplantibacillus plantarum

1

Lactobacillus rhamnosus

Lacticaseibacillus rhamnosus

1

Lactobacillus casei

Lacticaseibacillus casei

2
